# The Classification of Vestibular Schwannoma (VS) and Cerebellopontine Angle Meningioma (CPAM) Based on Multimodal Magnetic Resonance Imaging Analysis

**DOI:** 10.3390/diagnostics15091157

**Published:** 2025-05-01

**Authors:** Lihua Yuan, Jaming Lu, Xin Shu, Kun Liang, Cheng Wang, Jiu Chen, Zhishun Wang

**Affiliations:** 1Department of Interventional Radiography, Nanjing Drum Tower Hospital, Affiliated Hospital of Medical School, Nanjing University, Nanjing 210008, China; lihua.yuan@njglyy.com (L.Y.);; 2Department of Radiography, Nanjing Drum Tower Hospital, Affiliated Hospital of Medical School, Nanjing University, Nanjing 210008, China; 3Department of Psychiatry and Translational Imaging, Vagelos College of Physicians and Surgeons, Columbia University, New York, NY 10032, USA

**Keywords:** vestibular schwannoma (VSs), cerebellopontine angle meningioma (CPAMs), apparent diffusion coefficient (ADC), support vector machine (SVM), radiomics

## Abstract

**Background/Objectives**: This study evaluates the diagnostic efficacy of the apparent diffusion coefficient (ADC) and T1-weighted contrast-enhanced (T1W + C) and T2-weighted (T2W) imaging modalities in differentiating vestibular schwannomas (VSs) and cerebellopontine angle meningiomas (CPAMs), aiming to optimize clinical imaging protocols for these tumors. **Methods**: A retrospective analysis was conducted on 97 surgically and pathologically confirmed cases (65 VS, 32 CPAM) from Nanjing Drum Tower Hospital. Imaging features from ADC, T1W + C, and T2W sequences were extracted using medical imaging software. A support vector machine (SVM) model was trained to classify tumors based on these features, focusing on first-, second-, and third-order radiomic characteristics. **Results**: The ADC images demonstrated the highest classification efficiency, particularly with third-order features (AUC = 0.9817). The T2W images achieved the best accuracy (87.63%) using second-order features. Multimodal analysis revealed that ADC alone outperformed combinations with T1W + C or T2W sequences, suggesting limited added value from multi-sequence integration. **Conclusions**: Diffusion-weighted imaging (DWI) sequences, particularly ADC maps, exhibit superior diagnostic utility compared to T1W + C and T2W sequences in distinguishing VS and CPAM. The findings advocate prioritizing DWI in clinical imaging workflows to enhance diagnostic accuracy and streamline protocols.

## 1. Introduction

Brain tumors located at the cerebellopontine angle (CPA) are the most common neoplasms within the posterior fossa, comprising 5–10% of all brain tumors. Among all the types of CPA lesions, vestibular schwannomas (VSs) and CPA meningiomas (CPAMs) account for 70–80% and 10–15%, respectively [1]. Furthermore, research on IDH-mutant astrocytomas in the posterior cranial fossa (PCF) reveals that these tumors, although rare, pose significant diagnostic and therapeutic challenges due to their location and infiltrative nature. Multimodal treatment, including surgery, radiotherapy, and chemotherapy, is crucial for managing these tumors and achieving long-term disease control [2]. Trigeminal schwannomas (TSs), although rare, constitute a significant diagnostic and therapeutic challenge due to their complex anatomy and location, often requiring specialized surgical approaches such as an endoscopic transorbital approach (ETOA) for effective management. The endoscopic transorbital approach (ETOA) has emerged as a promising surgical option for treating trigeminal schwannomas (TSs), demonstrating high rates of gross total resection (GTR) and adaptability to individual patient needs [3].

Meningiomas originate from arachnoid villi, whereas most schwannomas arise from the vestibular nerves of the inner ear canal. In the CPA, meningiomas are the second most common tumors after VSs, including, in order of decreasing frequency, the posterior surface of the petrous bone, folds, slopes, cerebellar protrusions, and occipital holes. They often remain asymptomatic during growth, causing only spatial changes, particularly in CPA anatomy [4]. Meningiomas do not penetrate or erode the internal auditory canal (IAC), while VSs can penetrate the nerve bundles within the IAC [5]. Although the surgical management of vestibular schwannomas (VS) and cerebellopontine angle Meningiomas (CPAM) generally utilizes similar approaches—most commonly the retro sigmoid (posterior sigmoid suboccipital) approach—specific considerations such as hearing status and tumor extent can alter surgical strategy. For example, the trans labyrinthine approach is specifically indicated for VS patients in whom hearing preservation is no longer possible [6,7]. Therefore, accurately identifying these two tumors before surgery is critical to selecting the appropriate surgical procedures and treatment options. However, distinguishing between these two entities is challenging due to significant overlap [8]. Large trigeminal schwannomas, though benign, pose a significant surgical challenge due to their intricate location and often require a functional sparing approach to balance tumor removal with nerve preservation [9].

Other studies have found no significant differences in the clinical manifestations between CPAMs and schwannomas, indicating that these manifestations provide limited assistance in differentiating between the two. For example, regarding hearing abnormalities, there is no significant difference between patients with meningiomas and those with schwannomas [10]. The analysis of tumor samples on glass slides under a microscope, termed as histology, has been considered the golden standard for diagnosing brain tumors such as VS and CPAM. However, this method requires tissue removal, making it an invasive procedure. Thus, it serves as a final confirmatory diagnosis that patients choose reluctantly when no other options are available. Noninvasive brain imaging methods like magnetic resonance imaging (MRI) are employed for the routine examination of brain tumors, including VS and CPAM, to address this limitation. In some instances, VS and CPAM display distinct behaviors, which are evident as unique features in structural MRI data, aiding in their differentiation. For example, VS often involves the inner ear canal and presents heterogeneous tumor enhancement. In contrast, the ‘dural tail’ sign frequently appears in CPAM [11]. On the other hand, the features mentioned are generally absent in most cases, preventing the identification of VS and CPAM using this method for those applications. In practice, the structural MRI method has been expanded to include four types of pulse sequences: T2-weighted (T2w), T1-weighted (T1w), T1-weighted with enhancement (T1w+), and diffusion-weighted (DW) images (DWI). Among these four modalities, T1w+ images provide the most compelling diagnoses. However, acquiring such diagnoses often necessitates the injection of contrast media, which can be distressing to patients and may lead to contrast media sensitivity reactions.

The T2-weighted sequence is frequently utilized for diagnosing conditions related to the cerebellopontine angle, primarily indicating defects in the cerebrospinal fluid within the cerebellopontine cistern. A T1-weighted sequence with contrast is often employed to refine the diagnosis, necessitating the injection of a contrast agent. Reports concerning gadolinium-based contrast agents have recently received attention. Studies indicate that gadolinium can deposit multiple times in patients’ brains [12]. Although there is no evidence to show whether the deposition in the brain has a significant effect on brain function, the injection of GBCA is crucial for patients’ safety. A clinical allergy to GBCA is also common. Therefore, in clinical work, if we can reduce the injection of the invasive contrast agent, patients will be better protected.

Previous research in the region of the Antoni B cells has explored the role of diffusion-weighted MRI (DWI) with quantitative apparent diffusion coefficients (ADCs) in distinguishing between VSs and CPAMs [13]. VSs have a lower density, and their average ADC value is significantly higher than that of CPAMs. Studies mentioned a significant overlap between VS and CPAM, limiting the clinical application for tumor differentiation among individuals. Previously, obtaining an average ADC value entailed manually localizing the region of interest (ROI) on the tumor, overlooking its heterogeneity and potentially causing an overlap in average ADC values observed between the two entities [14,15]. The ADC map of the whole tumor is better than the localized approach that was used before because it gives more information about how heterogeneous the tumor is overall and its histological features [16]. Regions within the tumor exhibiting different diffusion characteristics are represented in the histogram when the entire tumor is analyzed. Histogram analysis using the complete tumor ROI method demonstrates advantages in distinguishing or classifying tumors and predicting therapeutic outcomes across various organs [17]. Earlier research assessed the efficacy of ADC maps, utilizing the full tumor ROI method combined with histogram analysis, in distinguishing radiologically ambiguous CPAM from VS. However, this research merely contrasted percentile differences between CPAM and VS [18]. They neglected to explore other potential numerical attributes derived from the ADC maps that could aid in identifying CPAM and VS. More importantly, they did not compare the performance of ADC maps to the image features acquired using other modalities of MRI that have been extensively used in clinics, such as T2w and T1w+ as described above, leaving the important question of how to objectively select these sequences in clinical applications unanswered. Therefore, in this study, for the first time, we systematically and comprehensively investigated the applicability of all the 19 possible numerical features computed from ADC maps and compared their performance with the same features computed from the other two clinically often used MRI modalities with the two pulse sequences, T2w and T1w+, in the identification and differentiation of CPAM and VS. We found that this significantly outperformed the other two sequences (T2w and T1w+) in most of these numerical features. Then, we further cross-validated the differentiation performance of CPAM and VS using ADC maps and their numerical features using machine learning based on a support vector machine (SVM).

## 2. Materials and Methods

### 2.1. Research Objectives

This study is a retrospective analysis. We collected data on meningiomas and vestibular schwannomas from the image archiving and communication system (PACS) of Nanjing Drum Tower Hospital, spanning August 2016 to March 2023. The dataset comprised 199 meningiomas and 74 vestibular schwannomas. From these, 32 patients with cerebellopontine angle meningiomas met our quality control criteria. Regarding the 74 vestibular schwannomas, 4 cases were excluded due to recurrence, as were another 5 because they lacked a DWI sequence; the remaining images satisfied our quality standards. Consequently, 65 cases were included in the study.

The inclusion criteria were as follows:Tumors located in the cerebellopontine region.Complete brain MRI images, encompassing at least T1W, T2W, DWI, and T1W + C sequences.MRI images meeting quality control standards.The exclusion criteria were as follows:Incomplete image sequences.Postoperative recurrent images.Presence of artifacts in MRI images, such as motion or metal artifacts.

Finally, we obtained 97 MR images of meningiomas and vestibular schwannomas in the cerebellopontine angle. There were 65 VS cases and 32 CPAM cases. We matched the two groups based on age (t = 1.10, *p* = 0.32; *t*-test) and gender (chi-square = 0.7, *p* = 0.26; chi-square test). Table 1 describes the characteristics of the participants.

### 2.2. Research Methods

#### 2.2.1. MRI Image Acquisition

A total of 5 MRI devices were included in this study. The specific parameters are listed below (Table 2a,b).

#### 2.2.2. Sketch of Interest Area

A senior radiologist standardized the DWI (diffusion-weighted imaging) sequence for two types of tumors, given that our images consist of only 18 layers. Subsequently, we calculated the apparent diffusion coefficient (ADC) values and generated an ADC map. To analyze the tumor, senior doctors outlined the tumor’s extent on each layer and retained the images of these layers. In the process of histogram-based ADC data analysis, we draw a Region of Interest (ROI) encompassing the entire tumor on each section of the ADC map, excluding areas with bleeding and adjacent blood vessels. The boundaries of the tumor are delineated using T2-weighted and contrast-enhanced T1-weighted images. We ensured that the ROI is marginally smaller than the actual tumor size to minimize the partial volume effect. Both the contrast-enhanced T1-weighted and T2-weighted images were carefully drawn. By cross-referencing other imaging sequences, we ascertained the precise location and borders of the tumor. Once the ROI in each region was defined, we computed the tumor volume by multiplying the sum of these areas by the slice thickness. Our primary focus in this study is the complete MRI representation of the tumor.

In addition to the DWI sequence, we compare the T1W enhancement and T2W sequences. Since the T1W enhancement sequence represents images taken after the injection of a contrast agent based on T1W, and given that T1W contains richer image information, we opt for the T1W enhancement sequence in our comparisons. This study aims to identify the optimal sequence or combination within the VS and CPAM image collections, providing crucial reference value for clinical applications.

#### 2.2.3. MATLAB Software Was Used to Generate a Total of 19 Imaging Features of Three Sequences -> MATLAB 2019b

These 19 imaging features include 11 first-order features (the first-order statistical features describe the distribution of voxel parameter values within the region of interest, usually based on histogram analysis): mean, median, standard deviation, entropy, 10th, 25th, 75th, 90th, 95th, 97th, and 100th percentile; five second-order features (the second-order statistic describes the spatial distribution complexity of lesions, also known as texture features): kurtosis, skewness, the third and fourth moments, and correlation dimension; and three third-order features (the third-order statistical features are further filtered): continuous wavelet transform (CWT) coefficients, power spectral density (PSD) using the Welch method, and periodogram. The imaging features of these latter three sequences are listed below (Table 3). CWT: The CWT is a time-localized wavelet decomposition tool that provides a time-frequency representation of signals, particularly effective for characterizing the changing features of non-stationary signals and identifying stationary segments within data streams, despite its high computational load and memory requirements. Pwelch: The pwelch algorithm estimates a signal’s power spectral density (PSD) using Welch’s method, which employs overlapping segment averaging and Hanning windowing to compute smoothed PSD estimates, making it suitable for the frequency-domain analysis of random signals. Periodogram: The periodogram is a PSD estimation method that identifies inherent periodic signals by evaluating the significance of frequencies in time-series data, especially useful for non-uniformly sampled datasets and optimizing the detection of periodic signal morphologies.

After extracting the imaging group features, 19 parameters of VS and CPAM patients were compared by *t*-test. A total of 19 parameters are used as the feature to train SVM model and classify them. The diagnostic efficiency of three sequences was evaluated by the characteristic curve of the subjects. The diagnostic efficiency was evaluated by the subjects curve combined with three sequences.

#### 2.2.4. Extract the Characteristics of Three Sequences of Imageology

In addition to analyzing the imaging characteristics of ADC, we compared T2W and T1W + C images using a *t*-test. For both T1W + C and T2W sequences, we employed five different feature combinations in SVM classification. We then compared these two groups and cross-classified the three most efficient feature groups from each sequence. Finally, we utilized an ROC curve to assess the classification efficiency.

#### 2.2.5. Statistical Method

We used a two-sample *t*-test to compare 19 parameters of VS and CPAM patients. Then, using a machine learning algorithm, SVM, we further studied the performance of these parameters in objectively classifying VS and CPAM.

SVM [19] is a method based on a hyperplane that separates classes. The principle of SVM is that all data bodies are regarded as different points distributed in multi-dimensional space (each feature of the data represents its dimension, and if there are n features, it forms an n-dimensional space). Through an algorithm, SVM finds a classification “hyperplane” (an N-1-dimensional subspace), optimizing the sample points to be classified and maximizing the minimum distance between sample points and the classification surface. SVM employs a Gaussian kernel function and uses the leave-one-out cross-validation (LOOCV) method [20]. In LOOCV, each sample serves as test data, while the remaining samples are used as training data to assess accuracy and generate receiver operating characteristic (ROC) curves.

Initially, we used the 19 parameters extracted from the patient’s ADC map as input features to train and validate the SVM model for VS and CPAM classification. Classification performance was evaluated using accuracy and ROC curves. Subsequently, based on T1W + C and T2W sequences, VS and CPAM were classified.

## 3. Results

### 3.1. The Results of Inter Group Differences in the Imaging Characteristics of ADC Images

Based on the ADC data from the DWI sequence, the mean values of the 19 parameters for CPAM and VS were compared using a *t*-test. Entropy, a first-order feature [21], could not be successfully extracted for most participants. Additionally, the other 10 first-order features indicated that the CPAM group was significantly larger than the VS group (Figure 1). In terms of second-order image features, kurtosis [22] and skewness [23] demonstrated that CPAM was significantly lower than VS, while the third and fourth moments were significantly greater for VS. Yet, there was no significant difference in the related dimension parameters between the two groups (Figure 2). Among the third-order image features, the CPAM group exhibited three significantly greater features: continuous wavelet transform coefficient [24], Welch power spectral density [25], and periodogram [26] when compared to the VS group (Figure 3).

### 3.2. The Difference Between Groups in the Imaging Characteristics of T2W

Based on T2w images, we tested the mean values of 19 parameters for both CPAM and VS using a *t*-test. While the first-order feature “entropy” could not be extracted for most subjects, the remaining first-order features, except for the 10th percentile, were not significant. The other nine features indicated that the CPAM group was significantly larger than the VS group (Figure 4).

Regarding the second-order image features, the kurtosis value for CPAM was significantly lower than that for VS. In addition, the related dimension parameters for the CPAM group were significantly larger compared to the VS group. However, there was no significant difference in the other three second-order dimensions: skewness and the third and fourth moments (Figure 5).

Concerning the third-order image features, the CPAM group exhibited significantly higher values in three features—continuous wavelet transform coefficient, Welch power spectral density, and periodogram—compared to the VS group (Figure 6).

### 3.3. The Difference Between Groups in the Imaging Characteristics of T1W + C Sequence

On the basis of the T1W sequence, a contrast agent containing gadolinium was injected during scanning. In normal structures, the meninges were clearly visible on enhanced sequences. In abnormal cases, contrast media with gadolinium can accumulate in the tumor parenchyma, highlighting the edges of the solid parts of the tumor and offering advantages in demonstrating tumor heterogeneity. In this study, one subject failed to complete the T1W enhancement sequence, so the sample size was 96 (CPAM n = 31, VS n = 65). Based on the images of the T1W enhancement sequence, we conducted a *t*-test on the mean values of 19 parameters of CPAM and VS. Among these, the first-order feature entropy could not be extracted for most subjects. For the remaining 10 first-order features, the VS group demonstrated a higher standard deviation (STD) compared to the CPAM group. Additionally, the CPAM group showed significantly higher values in nine other features compared to the VS group (Figure 7).

In the second-order image features, the kurtosis value of CPAM was significantly lower than that of VS. Furthermore, the correlation dimension and skewness of the CPAM group were significantly higher than those of the VS group, while the third and fourth moments showed no significant difference (Figure 8).

In the third-order image features, the CPAM group exhibited significantly lower values in three features—continuous wavelet transform coefficient, Welch power spectral density, and periodogram—compared to the VS group. However, there were more outliers present (Figure 9).

### 3.4. SVM Classification of CPAM and VS Based on Multimodal Characteristics

For the SVM classification of ADC images from DWI sequences, we employed five distinct feature combinations. The first group comprised all 18 image omics features, excluding the “entropy” parameter due to its unsuccessful extraction in most subjects. Next, the second group included all ten first-order features, the third group included five second-order features, the fourth group included three third-order features, and the fifth group included all 16 features that showed significant intergroup differences. Through these varied combinations, the SVM distinguished between CPAM and VS. The classification outcomes are presented in Table 4. Notably, aside from the second-order features, the efficacy of the other combinations reached commendable levels. Among them, the third-order features exhibited the highest accuracy, sensitivity, and specificity.

Further quantitative analysis using Receiver Operating Characteristic (ROC) curves revealed that the strategies employing first-order and third-order features achieved the highest Area Under the Curve (AUC) value of 0.9817. In contrast, the strategy based on second-order features had a significantly lower AUC of 0.8827 compared to the first- and third-order strategies (*p* < 0.05). However, no significant difference was observed between the first- and third-order feature strategies, both maintaining an AUC of 0.9817.

We classified SVM by using five different feature combinations for T1W + C and T2W, respectively. The first group includes all 18 imaging group features, except for those subjects who failed to extract the feature entropy parameter successfully; the second group includes all 10 first-order features; the third group includes all five second-order features; the fourth group includes all three third-order features; and the fifth group includes 14 features with significant group differences.

For the SVM classification results of T2W images, the results are shown in Table 5. It was found that the classification efficiency is the highest with an accuracy of 87.63%, and the classification performance of other strategies is close to that of all the features (ranging between 80.41% and 84.54%). However, in general, the classification accuracy of other classifiers is higher for the VS group with a larger number (both at 92.31%) and less accurate (all at 72% or less) for the CPAM group with fewer numbers. The results of the SVM classification of T1W + C images are similar to T2W; these results are shown in Table 6. Second-order features are used as classification features and the gender performance of SVM classification is optimal.

Further analysis of the ROC curve reveals that the second-order feature classifier outperforms others in terms of classification performance. Specifically, in the T2W image classifier, the AUC for the second-order feature is significantly higher than any other strategy. For the T1W + C image classifier, the AUC of the second-order feature surpasses both the first- and third-order feature classifiers. Moreover, it exceeds the AUC of classifiers using all features.

### 3.5. Comparison of the Imaging Group Characteristics of Three MRI Sequences

In order to evaluate the classification efficiency of three distinct sequences and the impact of different sequence feature combinations, we conducted a multimodal feature classification study. In particular, we selected the three most efficient feature groups from these sequences: the third-order features from the ADC image and the second-order features from both the T1W + C and T2W images. We then paired these features and also combined all three types. We performed the classification on data from 96 subjects with complete sequence information.

The results are presented in Table 7, which show that the ADC features alone—specifically, their third-order characteristics—are superior. Further analysis using ROC (Figure 10) indicates that the AUC for ADC classification is significantly higher than that for T1W + C, T2W, or any combination of these two. Combining ADC features with either T1W + C or T2W, or even combining features from all three sequences, did not significantly enhance classification performance compared to using ADC features alone.

## 4. Discussion

### 4.1. Multimodal Characteristics of VS and CPAM in Cerebellopontine Angle

Figure 11 illustrates the multimodal MRI features for the two tumor types and their respective sequences involved in this study. Both tumors are located in the cerebellopontine angle and exhibit no significant difference. On T2W imaging, both demonstrate moderate signal intensity, which is slightly uneven in the VS. Both of the T1W + C sequences show a high signal and clear enhancement. In the DWI sequence, they appear isointense but with slight variations in signal intensity: the CPAM signal is relatively uniform, whereas the VS lesion signal is less so. Clinically, this represents only one manifestation of what can be variable presentations for the same tumor type, thereby increasing diagnostic challenges [27].

### 4.2. ADC Imaging Characteristics of CPAM and VS in Cerebellopontine Angle

Previous studies on the differential diagnosis of ADC maps in these two diseases have simply compared individual histogram parameters without systematic comparison or multimodal analysis [18]. In our study, the CPAM group was significantly larger than the VS group in terms of first-order features. In terms of second-order image features, kurtosis and skewness in the CPAM group were significantly lower than those in the VS group, whereas the third and fourth moments in the CPAM group were significantly higher than those in the VS group. There was no significant difference in the related dimensions between the two groups. In terms of third-order image features, the CPAM group had significantly more features—continuous wavelet transform coefficient, Welch power spectral density, and periodogram—than the VS group. The CPAM group significantly outperforms the VS group, with the exception of kurtosis, skewness, and correlation dimension.

We carry out subsequent machine learning using an SVM on five different combinations of imaging features [28]. These combinations include the following: (1) the first group, which contains all 18 imaging features except “entropy”, which could not be successfully extracted for most subjects; (2) the second group, which includes all 10 first-order features; (3) the third group, which includes all five second-order features; (4) the fourth group, which includes all three third-order features; and (5) the fifth group, which includes all characteristics with significant differences between groups, totaling 16. These five groups of feature combinations were classified into the CPAM group and VS group. It was found that the third-order feature classification achieved the highest accuracy, sensitivity, and specificity [29]. Moreover, the ROC curve was used to quantitatively analyze these classification strategies. Both the first-order and third-order features had the highest AUC, at 0.9817 each.

The above research shows that machine learning enables the third-order features to achieve the highest classification accuracy for the two disease groups. Additionally, they exhibit the highest and most stable classification efficiency.

### 4.3. T2W Imaging Features of VS and CPAM in Cerebellopontine Angle

The *t*-test for the T2W images shows that there is no significant difference in the 10th percentile, third moment, fourth moment, and skewness, except for their entropy. The kurtosis in the CPAM group was significantly lower than in the VS group. The remaining 13 characteristics demonstrated that the CPAM group was significantly larger than the VS group. The image features of the T2W sequence are classified by SVM, and five different feature combinations are used for SVM classification: imaging features, first-order features, second-order features, third-order features, and features with significant intergroup differences. Using the second-order feature for classification yielded the highest efficiency at 87.63%.

### 4.4. T1W + C Imaging Features of VS and CPAM in Cerebellopontine Angle

The *t*-test on the T1W + C sequence images shows no significant difference between the third and fourth moments, except for the failure of entropy extraction. The standard deviation and peak value for the CPAM group were markedly lower than those for the VS group. In the meantime, the CPAM group significantly outperformed the VS group in 14 other characteristics. Support vector machine (SVM) was used to group image features from the T1W + C sequence into five different groups: all imaging histograms, all first-order features, all second-order features, all third-order features, and all features that showed significant intergroup differences. Mirroring the results seen with T2W, utilizing second-order features yielded the highest classification efficiency, achieving 81.25%.

### 4.5. Comparison of Multimodal Sequence Texture Analysis Features Between VS and CPAM in Cerebellopontine Angle

We used the three groups of features—third-order features of the ADC image and second-order features of T1W + C and T2W images—with the highest classification efficiency among the three sequences. We then paired these features with all three sequences. For classification, we utilized three sequences of case subjects. The ADC features exhibited the best classification performance. Moreover, ROC analysis revealed that the AUC of ADC classification was significantly higher than that of T1W + C, T2W, or a combination of T1W + C and T2W. However, combining ADC features with T1W + C or T2W, or all three sequences, did not significantly enhance the classification performance compared to using ADC features alone. In fact, these combinations did not even meet the performance standard of using only ADC features.

### 4.6. Limitations

In our study, we compared the classification accuracy of three sequences: apparent diffusion coefficient (ADC), T1-weighted with contrast (T1W + C), and T2-weighted (T2W) as calculated by diffusion-weighted imaging (DWI). The results revealed that the DWI sequence offers significant advantages in diagnosing both vestibular schwannoma (VS) and cerebellopontine angle meningioma (CPAM). During clinical scanning, emphasizing the importance of the DWI sequence and focusing on it can enhance efficiency. Normally, magnetic resonance imaging scans for patients are time-consuming. As the number of follow-up patients increases, appointment durations extend. To improve efficiency, technicians sometimes reduce image resolution or scanning layer thickness [30], which unfortunately can introduce unsatisfactory factors into the diagnosis. For instance, excessive thickness may lead to missing small lesions; if the thickness is set at 5 mm, lesions smaller than 5 mm might not be detected easily. Furthermore, lower image resolution also complicates the diagnostic process.

Recently, gadolinium-based contrast agents (GBCAs) have been reported to potentially deposit in the brain following repeated administration [31]. There is currently no evidence to determine the substantial impact of this deposition on brain function [32]. However, GBCA injections are crucial for patient safety. Allergic reactions to GBCAs are also common in clinical settings [33]. Thus, reducing the use of invasive contrast agents in clinical practice is vital for patient safety.

Our findings indicate that diffusion-weighted imaging (DWI) offers significant advantages in diagnosing cerebellopontine angle meningiomas (CPAMs) and vestibular schwannomas (VSs). In clinical workflows, focusing on DWI scans can enhance their quality and duration, potentially shortening other sequences or, in specific cases, conducting only a single DWI sequence to fulfill clinical diagnostic needs.

## 5. Conclusions

This study, based on the analysis of multimodal imaging groups, uses the traditional image group method to calculate image group characteristics. The identification of CPAM and VS involves combining different-order image group features to verify the two models in the verification set and evaluate the effectiveness of the prediction model. The results of this chapter provide an objective basis and quantitative evaluation method for imaging in identifying these two types of tumors, aiding clinical diagnosis. This radiomic investigation establishes ADC mapping as the most effective MRI biomarker for CPA tumor differentiation, achieving near-perfect discrimination (AUC > 0.98). These findings strongly support the revision of current imaging guidelines to emphasize DWI sequences in preoperative assessments.

## Figures and Tables

**Figure 1 diagnostics-15-01157-f001:**
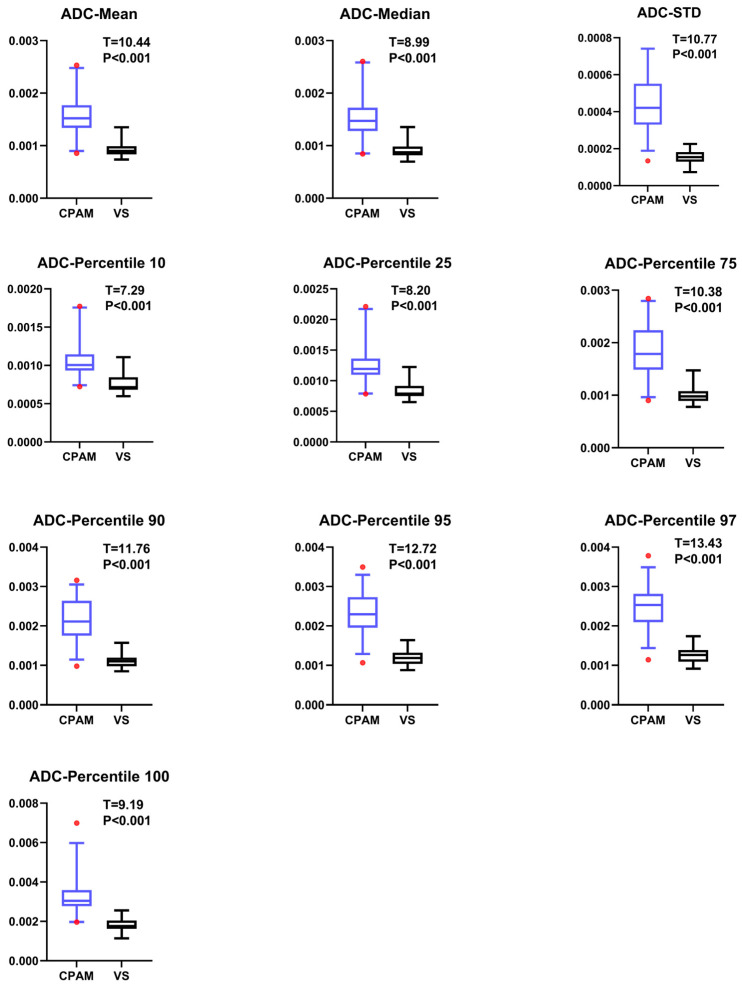
All images have CPAM on the left and VS on the right. The figure shows the group differences in the first-order image features of ADC between CPAM and VS groups, where 10 features showed that the CPAM group significantly exceeded the VS group.

**Figure 2 diagnostics-15-01157-f002:**
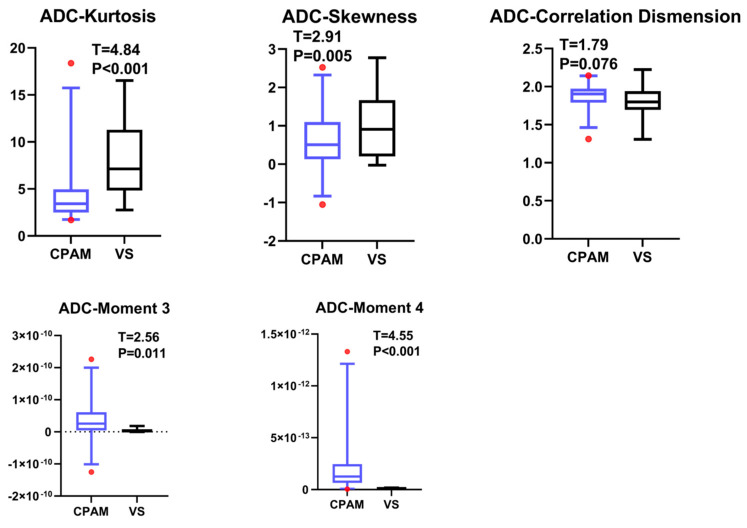
All images have CPAM on the left and VS on the right. The top graph shows the second-order image features of ADC between CPAM and VS groups. Both kurtosis and skewness for CPAM are less than VS, while torque CPAM is greater than VS for both the third and fourth moments. There is no significant difference in correlation dimension between the two groups.

**Figure 3 diagnostics-15-01157-f003:**
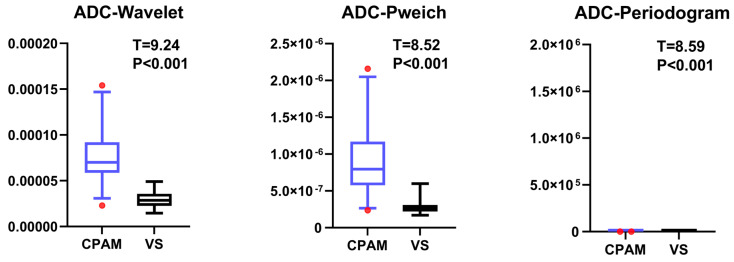
All images on the left are CPAM, while the right are VS. The upper figure shows the intergroup differences in the third-order image features of CPAM and VS patients, and all three features show significant differences in CPAM compared to VS.

**Figure 4 diagnostics-15-01157-f004:**
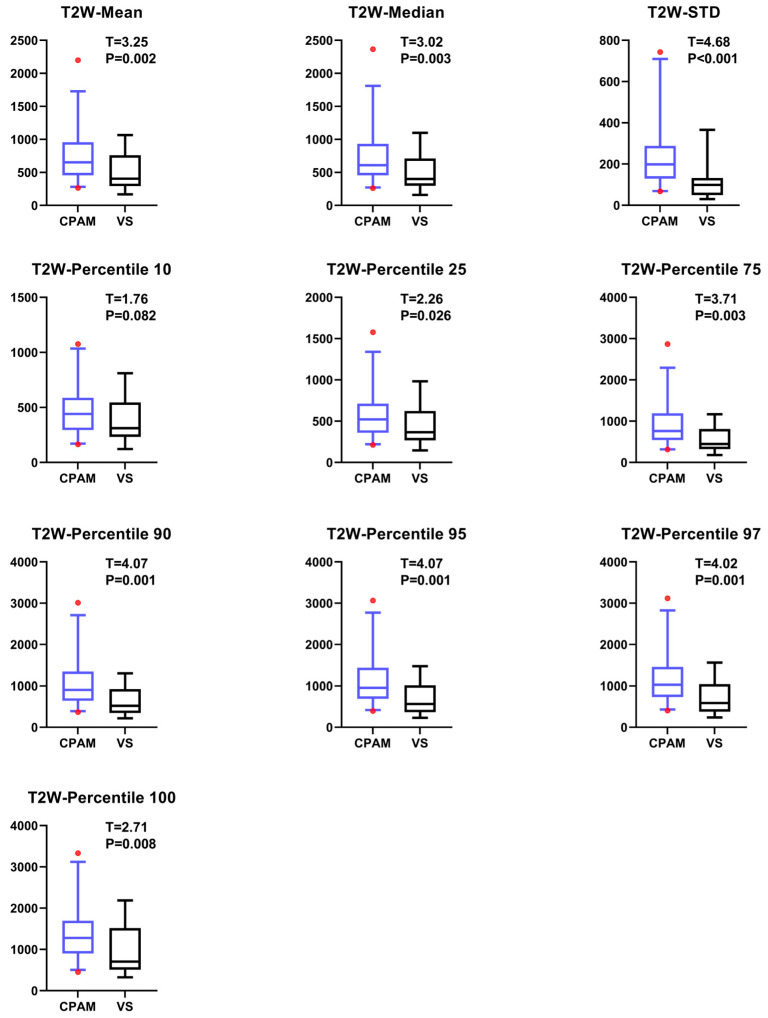
The intergroup differences in first-order image features between CPAM and VS patients in T2W sequence. The left side represents CPAM, and the right side represents VS. Except for the feature at the 10th percentile, which did not show a significant difference, all other features exhibited significant differences between the two groups.

**Figure 5 diagnostics-15-01157-f005:**
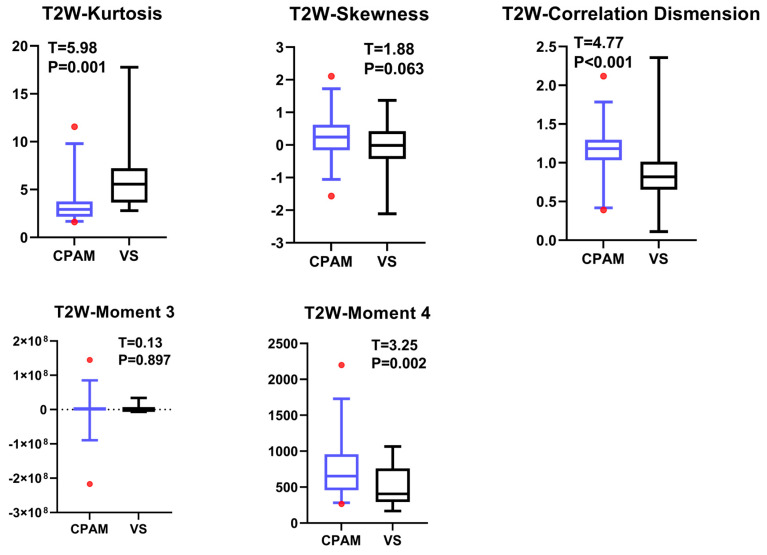
The intergroup differences in second-order image features between CPAM and VS patients in T2W sequence. The left side represents CPAM, and the right side represents VS. Only the correlation dimension exhibited intergroup differences.

**Figure 6 diagnostics-15-01157-f006:**
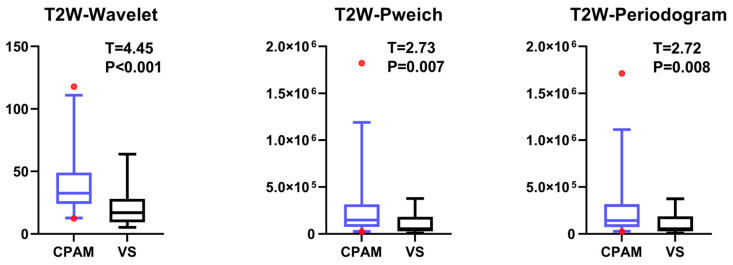
The third-order image features of T2W imaging in CPAM and VS patients show differences between the groups, with differences observed in three parameters, with the left side representing CPAM and the right side representing VS.

**Figure 7 diagnostics-15-01157-f007:**
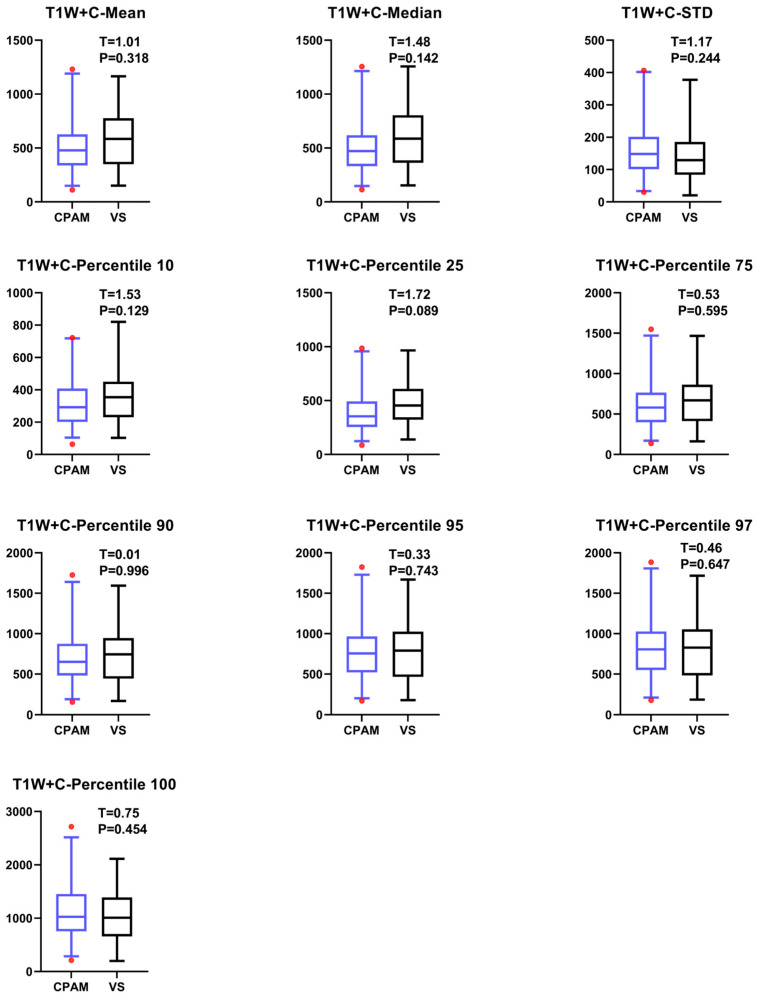
The T1W + C MRI images of CPAM and VS patients were presented in terms of their first-order image features. In addition to entropy, there were only 10 first-order features available. Among these, except for standard deviation (STD), which was significantly higher in VS group than in CPAM group, all other 9 features were significantly higher in CPAM group than in VS group.

**Figure 8 diagnostics-15-01157-f008:**
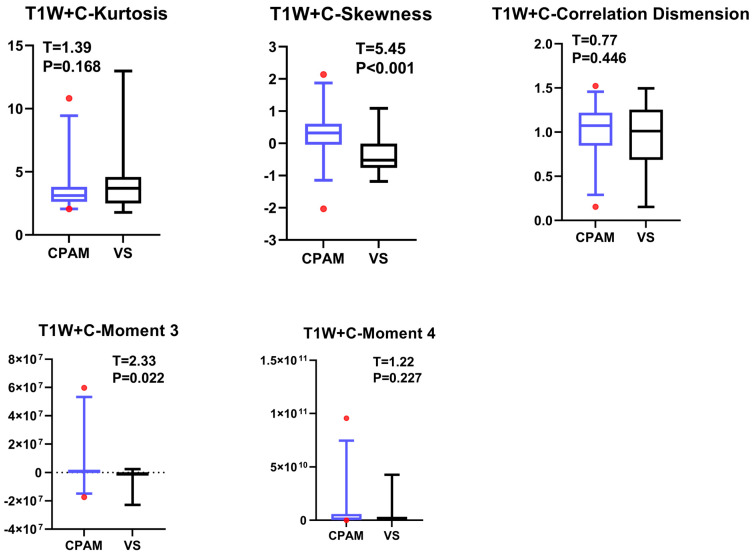
Taking T1W + C sequence images of CPAM and VS patients as the subject, the group differences in the second-order image features were investigated. It was found that the CPAM group showed significantly lower kurtosis values compared to the VS group, while related dimensions and skewness showed significantly higher values in the CPAM group compared to the VS group, and no significant differences were found in the third and fourth moments.

**Figure 9 diagnostics-15-01157-f009:**
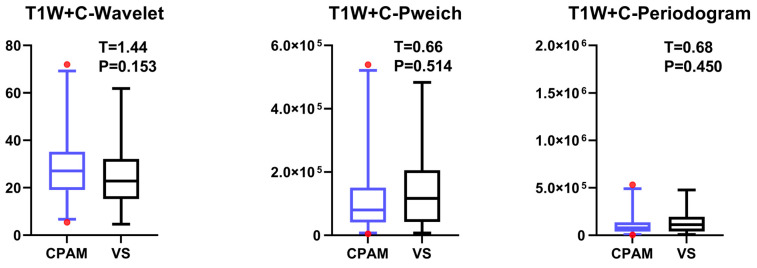
The T1W + C sequence images reveal significant intergroup differences in the third-order image features of CPAM and VS patients. All three features (wavelet, pweich, and periodogram) were significantly lower in the CPAM group compared to the VS group, but there were a considerable number of outliers present.

**Figure 10 diagnostics-15-01157-f010:**
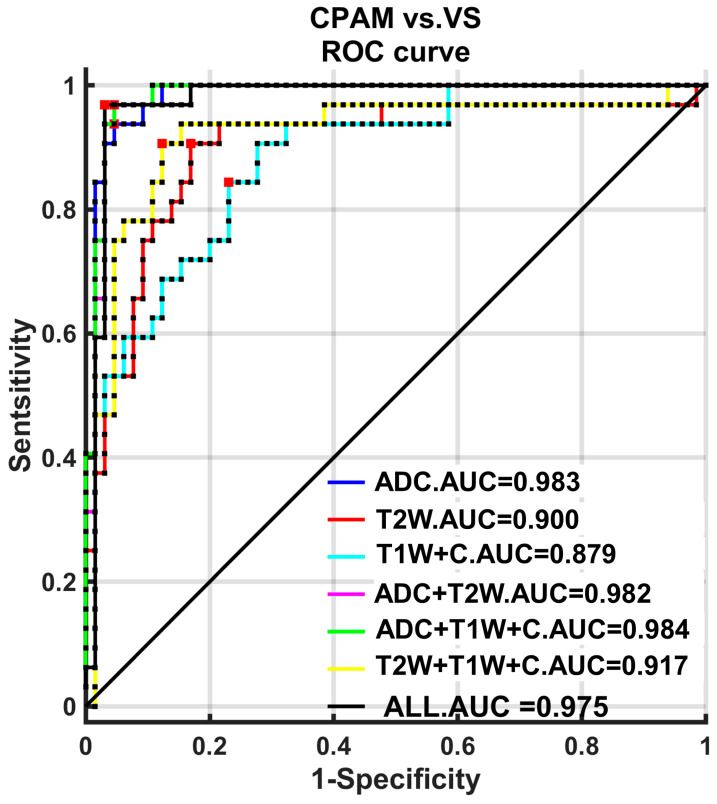
ROC curve based on multimodal sequence-based imaging genomics features for classification. ADC represents the three-order genomic feature classified with the best performance in the ADC sequence. T1W + C (T1W enhancement) and T2W are used for classification using the second-order features from these sequences. Addition represents their respective combinations, while ALL represents the imaging genomics features from all three sequences.

**Figure 11 diagnostics-15-01157-f011:**
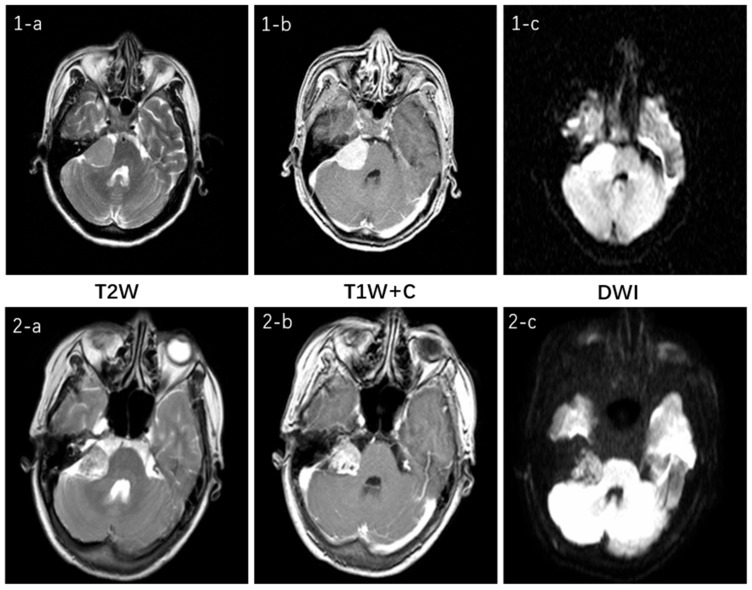
The top row, 1, represents CPAM, while the bottom row, 2, represents VS. a is the T2W sequence, b is the T1W +C sequence, and c is the DWI sequence. In the T2W sequence, they both show a medium signal with VS having a slightly uneven signal inside the tumor; in the T1W + C sequence, both show a high signal and obvious enhancement; in the DWI sequence, both are isointense, with slightly different signal intensity inside the lesion, although CPAM(1-c) shows a relatively uniform signal while VS(2-c) has a slightly uneven signal inside the lesion.

**Table 1 diagnostics-15-01157-t001:** Demographic information and tumor volume of CPAM and VS groups.

Characteristic	CPAM	VS	Differences Between Groups		
	N = 32	N = 65	Statistic	DF	*p*-Value
Age	54.8 ± 9.8	52.1 ± 11.4	t = 1.10	95	0.32
Gender	F = 21, M = 9	F = 41, M = 24	c² = 0.7	1	0.26
Tumor volume (mm^3^)	15.35 ± 19.52	16.81 ± 14.77	t = −0.41	1	0.68

**Table 2 diagnostics-15-01157-t002:** (**a**) Conventional scan parameters of MR of brain (Philips Achieva 1.5T). (**b**) Conventional scan parameters of MR of brain (Philips TX 3.0T).

(**a**)									
**Sequence**	**Tech**	**Ori**	**TR**	**TE**	**FA**	**Matrix**	**Thi**	**Gap**	**NSA**
T2W	TSE	TRA	3908	100	90	272 × 166	6	1	3
T1W	SE	TRA	113	157	80	244 × 184	6	1	3
DWI	SE	TRA	2062	67	90	176 × 107	6	1	2
T1W + C	SE	TRA	120	3.0	80	256 × 113	6	1	2
(**b**)									
**Sequence**	**Tech**	**Ori**	**TR**	**TE**	**FA**	**Matrix**	**Thi**	**Gap**	**NSA**
T2W	TSE	TRA	2332	80	90	208 × 120	6	1	1
T1W	SE	TRA	181	46	80	232 × 147	6	1	2
DWI	SE	TRA	2139	81	90	144 × 113	6	1	2
T1W + C	SE	TRA	132	2.1	80	328 × 260	6	1	2

**Table 3 diagnostics-15-01157-t003:** Image omics feature selection.

Computing Method	Omics Characteristics	Number
first-order features	mean, median, standard deviation, entropy, 10th, 25th, 75th, 90th, 95th, 97th, and 100th percentile	11
second-order features	kurtosis, skewness, the third and fourth moments, and correlation dimension	5
third-order features	continuous wavelet transform (CWT) coefficients, power spectral density (PSD) using Welch method, and periodogram	3

**Table 4 diagnostics-15-01157-t004:** Evaluation of SVM classifier performance using various combinations of radiomics features extracted from ADC maps.

	All Features	First-Order Feature	Second-Order Features	Third-Order Features	Significant Difference
Accuracy	94.85%	95.88%	79.38%	96.91%	95.88%
Sensitivity *	96.92%	96.92%	87.69%	96.92%	96.92%
Specificity *	90.63%	93.75%	62.50%	96.88%	93.75%

* Sensitivity and specificity were defined as positive in VS group.

**Table 5 diagnostics-15-01157-t005:** Performance of SVM classification based on different combinations of image omics features using T2W map.

	All Features	First-Order Feature	Second-Order Features	Third-Order Features	Significant Difference
Accuracy	84.54%	80.41%	87.63%	80.41%	81.44%
Sensitivity *	95.83%	92.31%	93.85%	93.85%	92.31%
Specificity *	62.50%	56.25%	75%	53.13%	59.38%

* Sensitivity and specificity were defined as positive in VS group.

**Table 6 diagnostics-15-01157-t006:** Performance of SVM classification based on T1W + C map with different combinations of image omics features.

	All Features	First-Order Feature	Second-Order Features	Third-Order Features	Significant Difference
Accuracy	80.21%	71.88%	81.25%	72.92%	78.13%
Sensitivity *	95.38%	100%	95.38%	100%	93.85%
Specificity *	48.39%	12.90%	51.60%	16.13%	45.16%

* Sensitivity and specificity were defined as positive in VS group.

**Table 7 diagnostics-15-01157-t007:** Multimodal classification performance.

	ADC	T2W	T1WC	ADC + T2W	ADC + T1WC	T1WC + T2W	ALL
Accuracy	96.88%	81.25%	81.25%	95.83%	94.79%	84.38%	92.71%
Sensitivity *	96.92%	92.31%	95.38%	96.92%	96.92%	92.31%	96.92%
Specificity *	96.77%	58.06%	51.61%	93.55%	90.32%	67.74%	83.87%

* Sensitivity and specificity were defined as positive in VS group.

## Data Availability

The original contributions presented in the study are included in the article, further inquiries can be directed to the corresponding author.

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
