# Peer review of "The Classification of Vestibular Schwannoma (VS) and Cerebellopontine Angle Meningioma (CPAM) Based on Multimodal Magnetic Resonance Imaging Analysis"

_diagnostics, 2025, doi:10.3390/diagnostics15091157_

Round 1
Reviewer 1 Report
Comments and Suggestions for Authors
Dear Author,
1- According to Paragraph 47,48,49 as has been mentioned, "Given their distinct characteristics, such as shapes and locations, these two types of tumors require different treatments. For example, the posterior sigmoid suboccipital approach is recommended for CPAMs, while the expanded trans- scarring approach is commonly used for VSs [4, 5]." is not correct. The surgical approach for both CPAM and VS is same in most of cases except in cases where hearing has been lost completely, translabirynthine approach can be used
2-Though your conclusions, strongly support revising current imaging guidelines to emphasize DWI sequences in preoperative assessments, there is no clear cut evidence negating the use of T1W +C which you find bothersome for patients.
Author Response
Comments 1: According to Paragraph 47,48,49 as has been mentioned, "Given their distinct characteristics, such as shapes and locations, these two types of tumors require different treatments. For example, the posterior sigmoid suboccipital approach is recommended for CPAMs, while the expanded trans- scarring approach is commonly used for VSs [4, 5]." is not correct. The surgical approach for both CPAM and VS is same in most of cases except in cases where hearing has been lost completely, trans labirynthine approach can be used.
Response 1: We deeply appreciate the reviewer's insightful comment. Indeed, upon closer examination, it is evident that Vestibular Schwannomas (VS) and Cerebellopontine Angle Meningiomas (CPAM) are typically managed using comparable surgical approaches, notably the retro sigmoid (posterior sigmoid suboccipital) approach, in clinical practice. However, there is a selective preference for the trans-labyrinthine approach in VS cases where hearing preservation is no longer a viable option, owing to its direct pathway and superior exposure. Therefore, accurate preoperative differentiation remains paramount, guiding delicate surgical decisions concerning hearing preservation, the extent of resection, and customized intraoperative strategies. The corresponding adjustments, located on lines 57-62 of the second page, have been incorporated into the revised manuscript and have been updated accordingly.
Comments 2: Though your conclusions, strongly support revising current imaging guidelines to emphasize DWI sequences in preoperative assessments, there is no clear cut evidence negating the use of T1W +C which you find bothersome for patients.
Response 2: We extend our gratitude for your meticulous evaluation. It is acknowledged that T1W+C, derived from T1W, carries a wealth of information. The three sequences compared in our article are: T1W+C, T2W, and DWI. We acknowledge our error in labeling the figure and have rectified it by changing T1W to T1W+C. This correction ensures clarity and consistency throughout the manuscript.
Reviewer 2 Report
Comments and Suggestions for Authors
The authors conducted a retrospective study analyzing MRI sequences—ADC, T1-weighted with contrast (T1W+C), and T2-weighted (T2W)—to differentiate vestibular schwannoma (VS) from cerebellopontine angle meningioma (CPAM). Radiomic features were extracted from each modality, and support vector machine (SVM) models were trained to evaluate classification performance. ADC-derived third-order features achieved the highest diagnostic accuracy (AUC = 0.9817), outperforming both individual and combined modalities
1)The use of APC in the study of swhannomas obviously applies to those of the V cranial nerve as well; you might expand the introduction by reading and citing these two seminal studies
https://doi.org/10.1007/s00701-024-06292-8
https://doi.org/10.3390/jcm13133701
2)he paper refers to “first-order”, “second-order”, and “third-order” features, but these are not clearly defined in terms of image processing or radiomic conventions. Please include a table or section clarifying what is meant by each order, especially the inclusion of features like Welch power spectral density under “third-order”
3)The conclusion that ADC alone is superior is important, but the relatively lower performance of multimodal combinations needs deeper discussion. Were there issues with feature redundancy or overfitting?
4) Posterior fossa tumors can always lead to diagnostic complexities see for example this very recent article that I suggest reading and cite: https://doi.org/10.1007/s10143-025-03436-x
5)Figure legends should be more descriptive—explain what each feature represents (e.g., "wavelet", "pwelch"). Tables 3 to 6 are well-structured but could benefit from standard deviation or confidence intervals.
Dear authors, I am looking forward to reading revised manuscript
Comments on the Quality of English Language
I suggest minor editin
A careful proofreading is needed. Examples:
-
“We found that significantly outperformed the other two sequences…” (missing subject)
-
“...most of these of these numerical features…” (repetition)..
Author Response
Comments 1: The use of APC in the study of swhannomas obviously applies to those of the V cranial nerve as well; you might expand the introduction by reading and citing these two seminal studies
https://doi.org/10.1007/s00701-024-06292-8
https://doi.org/10.3390/jcm13133701
Response 1: Thank you for the recommended references. Upon thorough examination, I have found them immensely useful. I have incorporated the essence of both studies into lines 44-50 and 67-69 of the paper and added them to the bibliography as references 3 and 9, respectively..
Comments 2: the paper refers to “first-order”, “second-order”, and “third-order” features, but these are not clearly defined in terms of image processing or radiomic conventions. Please include a table or section clarifying what is meant by each order, especially the inclusion of features like Welch power spectral density under “third-order”
Response 2: Thank you for your insightful feedback. To enhance clarity, we have introduced a new paragraph 2.2.3. (lines 185-205) detailing the three types of features. Specifically, regarding third-order features. The details are as follows: It includes 11 first-order features(The first-order statistical features describe the dis-tribution of voxel parameter values within the region of interest, usually based on histo-gram analysis): Mean, median, standard deviation, entropy, 10th, 25th, 75th, 90th, 95th, 97th and 100th percentile; Five second-order features(The second-order statistic describes the spatial distribution complexity of lesions, also known as texture features): kurtosis, skewness, the third and fourth moments and correlation dimension; Three third-order features(The third-order statistical features are further filtered): continuous wavelet trans-form (CWT) coefficients, power spectral density (PSD) using Welch method and periodo-gram. The imaging features of three sequences are listed below (tables 3). CWT: The CWT is a time-localized wavelet decomposition tool that provides a time-frequency representation of signals, particularly effective for characterizing the changing features of non-stationary signals and identifying stationary segments within data streams, despite its high computational load and memory requirements. Pwelch: The pwelch algorithm estimates a signal's power spectral density (PSD) using Welch's method, which employs overlapping segment averaging and Hanning windowing to compute smoothed PSD estimates, making it suitable for frequency-domain analysis of random signals. Periodogram: The periodogram is a PSD estimation method that identifies inherent peri-odic signals by evaluating the significance of frequencies in time-series data, especially useful for non-uniformly sampled datasets and optimizing detection of periodic signal morphologies.
Comments 3: The conclusion that ADC alone is superior is important, but the relatively lower performance of multimodal combinations needs deeper discussion. Were there issues with feature redundancy or overfitting?
Response 3: The conclusion that ADC alone performs superiorly is noteworthy. However, the relatively lower performance of multimodal combinations merits further discussion. The increased feature space associated with multimodal combinations complicates the model and elevates the risk of overfitting. With a limited dataset of 97 confirmed cases (65 vs., 32 CPAM), complex models trained on such data are prone to overfitting, performing well on training data but potentially poorly on unseen data. Combining ADC with T1W+C, T2W, or all sequences did not significantly enhance performance compared to ADC alone, sometimes even resulting in slight performance degradation. This suggests minimal added value from other sequences' features. The study emphasizes the importance of selecting informative and non-redundant features for accurate tumor classification. Future research should optimize feature selection and model complexity, increase dataset size, and validate on untrained data to maximize multimodal imaging benefits while minimizing redundancy and overfitting risks.
Comments 4: Posterior fossa tumors can always lead to diagnostic complexities see for example this very recent article that I suggest reading and cite: https://doi.org/10.1007/s10143-025-03436-x
Response 4: Thank you for your references. We have thoroughly reviewed it and found it to be extremely beneficial. A summary of this paper has been included in lines 40-44 of the paper, and the reference has been added (reference 2).
Comments 5: Figure legends should be more descriptive—explain what each feature represents (e.g., "wavelet", "pwelch"). Tables 3 to 6 are well-structured but could benefit from standard deviation or confidence intervals.
Response 5: The article added Table 3 (line 211) to classify the three types of features in a table. Tables 4 to 7 are well structured and do benefit from standard deviations or confidence intervals. The deficiencies in this regard will be improved in future work.
Round 2
Reviewer 2 Report
Comments and Suggestions for Authors
Fine, all isse addressed